# Augmenting Transformers with Recursively Composed Multi-grained Representations

**Xiang Hu**[1], **Qingyang Zhu**[2], **Kewei Tu**[2*], **Wei Wu**[1*]

[1]Ant Group [2]ShanghaiTech University

{aaron.hx; congyue.ww}@antgroup.com;
{zhuqy; tukw}@shanghaitech.edu.cn

## Abstract

We present ReCAT, a recursive composition augmented Transformer that is able to explicitly model hierarchical syntactic structures of raw texts without relying on gold trees during both learning and inference. Existing research along this line restricts data to follow a hierarchical tree structure and thus lacks inter-span communications. To overcome the problem, we propose novel contextual inside-outside (CIO) layers, each of which consists of a top-down pass that forms representations of high-level spans by composing low-level spans, and a bottom-up pass that combines information inside and outside a span. The bottom-up and top-down passes are performed iteratively by stacking CIO layers to fully contextualize span representations. By inserting the stacked CIO layers between the embedding layer and the attention layers in Transformer, the ReCAT model can perform both deep intra-span and deep inter-span interactions, and thus generate multi-grained representations fully contextualized with other spans. Moreover, the CIO layers can be jointly pre-trained with Transformers, making ReCAT enjoy scaling ability, strong performance, and interpretability at the same time. We conduct experiments on various sentence-level and span-level tasks. Evaluation results indicate that ReCAT can significantly outperform vanilla Transformer models on all span-level tasks and recursive models on natural language inference tasks. More interestingly, the hierarchical structures induced by ReCAT exhibit strong consistency with human-annotated syntactic trees, indicating good interpretability brought by the CIO layers. [1]

## 1 Introduction

Recent years have witnessed a plethora of breakthroughs in the field of natural language processing (NLP), thanks to the advances of deep neural techniques such as Transformer (Vaswani et al., 2017), BERT (Devlin et al., 2019), and GPTs (Brown et al., 2020; OpenAI, 2023). Despite the success of the architecture, syntax and semantics in Transformer models are represented in an implicit and entangled form, which is somewhat divergent from the desiderata of linguistics. From a linguistic point of view, the means to understand natural language should comply with the principle that "the meaning of a whole is a function of the meanings of the parts and of the way they are syntactically combined" (Partee, 1995). Hence, it is preferable to model hierarchical structures of natural language in an explicit fashion. Indeed, explicit structure modeling could enhance interpretability (Hu et al., 2023), and result in better compositional generalization (Sartran et al., 2022). The problem is then how to combine the ideas of Transformers and explicit hierarchical structure modeling, so as to obtain a model that has the best of both worlds.

In this work, we attempt to answer the question by proposing novel **C**ontextual **I**nside-**O**utside (CIO) layers as an augmentation to the Transformer architecture and name the model "Recursive Composition Augmented Transformer" (ReCAT). Figure 1 illustrates the architecture of ReCAT. In a nutshell, ReCAT stacks several CIO layers between the embedding layer and the deep self-attention layers in Transformer. These layers explicitly emulate the hierarchical composition process of language and

---

*Corresponding authors

[1]Code released at `https://github.com/ant-research/StructuredLM_RTDT`.

provide Transformers with explicit multi-grained representations. Specifically, we propose a variant of the deep inside-outside algorithm (Drozdov et al., 2019), which serves as the backbone of the CIO layer. As shown in Figure 1(a), multiple CIO layers are stacked to achieve an iterative up-and-down mechanism to learn contextualized span representation. During the bottom-up pass, the model composes low-level constituents to refine high-level constituent representations from the previous iteration, searching for better underlying structures via a dynamic programming approach (Maillard et al., 2017); while during the top-down pass, each span merges information from itself, its siblings, and its parents to obtain a contextualized representation. By this means, the stacked CIO layers are able to explicitly model hierarchical syntactic compositions of inputs and provide contextualized constituent representations to the subsequent Transformer layers, where constituents at different levels can sufficiently communicate with each other via the self-attention mechanism. ReCAT enjoys several advantages over existing methods: first, unlike Transformer Grammars (Sartran et al., 2022), our model gets rid of the dependence on expensive parse tree annotations from human experts, and is able to recover hierarchical syntactic structures in an unsupervised manner; second, unlike DIORA (Drozdov et al., 2019) and R2D2 (Hu et al., 2021), our model breaks the restriction that information exchange only happens either inside or outside a span and never across the span boundaries, and realizes cross-span communication in a scalable way. Moreover, we reduce the space complexity of the deep inside-outside algorithm from cubic to linear and parallel time complexity to approximately logarithmic. Thus, the CIO layers can go sufficiently deep, and the whole ReCAT model can be pre-trained with vast amounts of data.

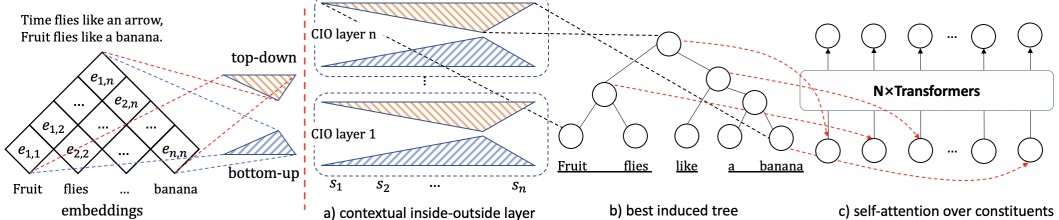

Figure 1: ReCAT model architecture. The contextual inside-outside layers take $n$ token embeddings and output $2n - 1$ node representations.

We evaluate ReCAT on GLUE (Wang et al., 2019) (sentence-level tasks), OntoNotes (Weischedel et al., 2013) (span-level tasks), and PTB (Marcus et al., 1993) (grammar induction). The empirical results indicate that: (1) ReCAT attains superior performance compared to Transformer-only baselines on all span-level tasks with an average improvement over 3%; (2) ReCAT surpasses the hybrid models composed of RvNNs and Transformers by 5% on natural language inference; and (3) ReCAT outperforms previous RvNN based baselines by 8% on grammar induction. Our main contributions are three-fold:

1. We propose a Contextual Inside-Outside (CIO) layer, which can be jointly pre-trained with Transformers efficiently, induce underlying syntactic structures of text, and learn contextualized multi-grained representations.

2. We further propose ReCAT, a novel architecture that achieves direct communication among constituents at different levels by combining CIO layers and Transformers.

3. We reduce the complexity of the deep inside-outside algorithm from cubic to linear, and further reduce the parallel time complexity from linear to approximately logarithmic.

## 2 RELATED WORKS

The idea of equipping deep neural architectures with syntactic structures has been explored in many studies. Depending on whether gold parse trees are required, existing work can be categorized into two groups. In the first group, gold trees are part of the hypothesis of the methods. For example, Pollack (1990) proposes encoding the hierarchical structures of text with an RvNN; Socher et al. (2013) verify the effectiveness of RvNNs with gold trees for sentiment analysis; RNNG (Dyer et al., 2016) and Transformer Grammars (Sartran et al., 2022) perform text generation with syntactic knowledge. Though rendering promising results, these works suffer from scaling issues due to the expensive and elusive nature of linguistic knowledge from human experts. To overcome the challenge, the other group of work induces the underlying syntactic structures in an unsupervised manner. For instance, ON-LSTM Shen et al. (2019b) and StructFormer (Shen et al., 2021) integrate syntactic struc-

tures into LSTMs or Transformers by masking information in differentiable ways; Yogatama et al. (2017) propose jointly training the shift-reduce parser and the sentence embedding components; URNNG (Kim et al., 2019b) applies variational inference over latent trees to perform unsupervised optimization of the RNNG; Chowdhury & Caragea (2023) propose a Gumbel-Tree (Choi et al., 2018) based approach that can induce tree structures and cooperate with Transformers as a flexible module by feeding contextualized terminal nodes into Transformers; Maillard et al. (2017) propose an alternative approach based on a differentiable neural inside algorithm; Drozdov et al. (2019) propose DIORA, an auto-encoder-like pre-trained model based on an inside-outside algorithm (Baker, 1979; Casacuberta, 1994); and R2D2 reduces the complexity of the neural inside process by pruning, which facilitates exploitation of complex composition functions and pre-training with large corpus. Our work is inspired by DIORA and R2D2, but a major innovation of our work is that our model can be jointly pre-trained with Transformers efficiently, which is not trivial to achieve for DIORA and R2D2. Both of them have to obey the information access constraint (i.e., information flow is only allowed within (inside) or among (outside) spans), due to their training objectives of predicting each token via its context span representations. Joint pre-training with Transformers would break the constraint and lead to information leakage, thus not feasible.

# 3 METHODOLOGY

## 3.1 MODEL

Overall, there are two basic components in ReCAT: CIO layers and Transformer layers. As illustrated in Figure 1, the CIO layers take token embeddings as inputs and output binary trees with node representations. During the iterative up-and-down encoding within the CIO layers, each span representation learns to encode both context and relative positional information. The multi-grained representations are then fed into the Transformer layers where constituents at different levels can interact with each other directly.

### 3.1.1 PRUNING ALGORITHM

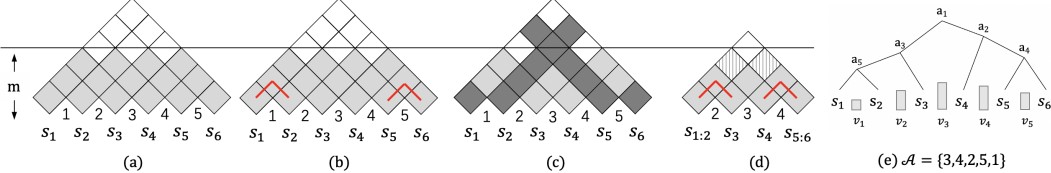

Figure 2: Example of the pruning process. $s_i$ and $s_{i:j}$ denotes the $i$-th token and sub-string covering $i$ to $j$, both included. The numbers are split indices. In (e), the merge order is the reverse order of the split order, which is $\{1, 5, 2, 4, 3\}$

We propose to improve the pruned deep inside algorithm of Hu et al. (2022) so that it can be completed in approximately logarithmic steps. The main idea is to prune out unnecessary cells and simultaneously encode the remaining cells once their sub-span representations are ready. As shown in Figure 2(e), an unsupervised parser is employed to assign each split point a score and recursively splits the sentence in the descending order of the scores. Thus the reverse of the split order can be considered as the merge order of spans and cells that would break the merged spans are deemed unnecessary. During the pruning pass, we record the encoding order $\mathcal{B}$ of cells and their valid splits $\mathcal{K}$. As shown in Figure 2, there is a pruning threshold $m$ which is triggered when the height of encoded cells (in grey) in the chart table reaches $m$. At the beginning, cells beneath $m$ are added bottom-up to $\mathcal{B}$ row by row. Once triggered, the pruning process (refer to (a),(b),(c),(d) in Figure 2) works as follows:

0. Group split points according to their height in the tree, with each group corresponding to their height in the induced tree, e.g., group1: $\{1, 5\}$, group2: $\{2, 4\}$, group3: $\{3\}$.
1. Merge all spans simultaneously in the current merge group, e.g., $(s_1, s_2)$ and $(s_5, s_6)$ in (b).
2. Remove all conflicting cells that would break the now non-splittable span from Step 1, e.g., the dark cells in (c), and reorganize the chart table much like in the Tetris game as in (d).
3. Append the cells that just descend to height $m$ to $\mathcal{B}$ and record their valid splits in $\mathcal{K}$, e.g., the cell highlighted with stripes in (d) whose valid splits are $\{2, 3\}$ and $\{3, 4\}$ respectively. Then go back to Step 1 until no cells are left.

In this way, the entire inside algorithm can be finished within tree-height steps, whose parallel time complexity is approximately logarithmic based on our empirical study in Appendix A.6. The detailed pseudo-code can be found in Appendix A.5. The optimization of the unsupervised parser is explained in the next section.

### 3.1.2 Contextual Inside-outside layers

In this part, we explain contextual inside-outside layers, and how to extend linear neural inside algorithm to our contextual inside-outside layers.

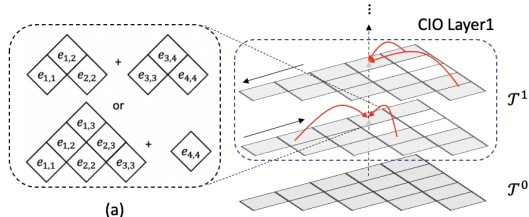

Figure 3: The first layer of stacked CIO layers.

Our contextual inside-outside algorithm borrows the neural inside-outside idea from DIORA (Drozdov et al., 2019), but differs from it in the following ways. First, the outside pass is designed to contextualize span representation instead of constructing auto-encoding loss. As a result of introducing information leakage, the original pre-training objective does not work anymore. Second, we introduce an iterative up-and-down mechanism by stacking multiple CIO layers to refine span representations and underlying structures layer by layer. We use masked language modeling as the new pre-training objective thanks to the mechanism allowing spans affected by masked tokens to contextualize their representations through iterative up-and-down processes.

**Denotations.** Given a sentence $\mathbf{S} = \{s_1, s_2, ..., s_n\}$ with $n$ tokens, Figure 3 shows multi-layered chart tables. We denote the chart table at the $l$-th layer as $\mathcal{T}^l$, where each cell $\mathcal{T}^l_{i,j}$ covers span $(i, j)$ corresponding to sub-string $s_{i:j}$. For span $(i, j)$, its set of **inside pair** consists of span pairs $\{(i, k), (k + 1, j)\}$ for $k = i, ..., j - 1$. For span $(i, j)$, its set of **outside pair** consists of span pairs $\{(i, k), (j + 1, k)\}$ for $k = j + 1, ..., n$ and $\{(k, j), (k, i - 1)\}$ for $k = 1, ..., i - 1$. For example, given a sentence $\{s_1, s_2, s_3, s_4\}$, the inside pairs of span $(1, 3)$ are $\{(1, 1), (2, 3)\}$ and $\{(1, 2), (3, 3)\}$, and the outside pairs of span $(2, 3)$ are $\{(1, 3), (1, 1)\}$ and $\{(2, 4), (4, 4)\}$. We denote the representations computed during the inside pass and the outside pass as $\hat{e}$ and $\check{e}$ respectively.

**Pruned Inside Pass.** During the inside process at the $l$-th layer, given an inside pair of a span $(i, j)$ split on $k$ s.t. $i \leq k < j$, we denote its composition representation and compatibility score as $\hat{e}^l_{i,j}[k]$ and $a^l_{i,j}[k]$ respectively. The compatibility score means how likely an inside pair is to be merged. $\hat{e}^l_{i,j}[k]$ is computed by applying the composition function to its immediate sub-span representations $\hat{e}^l_{i,k}$ and $\hat{e}^l_{k+1,j}$. Meanwhile, the composition function additionally takes the outside representation $\check{e}^{l-1}_{i,j}$ from the previous layer to refine the representation, as illustrated in Figure 3. $a^l_{i,j}[k]$ is computed by recursively summing up the scores of the single step compatibility score $\bar{a}^l_{i,j}[k]$ and the corresponding immediate sub-spans, i.e., $a^l_{i,k}$ and $a^l_{k+1,j}$. The inside score $a^l_{i,j}$ and representation $\hat{e}^l_{i,j}$ of each cell $\mathcal{T}_{i,j}$ are computed using a soft weighting over all possible inside pairs, i.e., $a^l_{i,j}[k]$ and $\hat{e}^l_{i,j}[k]$ with $k \in \mathcal{K}_{i,j}$, where $\mathcal{K}$ is the set of valid splits for each span generated during the pruning stage. The inside representation and score of a cell are computed as follows:

$$\hat{e}^l_{i,j}[k] = \text{Compose}^l_\alpha(\hat{e}^l_{i,k}, \hat{e}^l_{k+1,j}, \check{e}^{l-1}_{i,j}), \bar{a}^l_{i,j}[k] = \phi_\alpha(\hat{e}^l_{i,k}, \hat{e}^l_{k+1,j}), a^l_{i,j}[k] = \bar{a}^l_{i,j}[k] + a^l_{i,k} + a^l_{k+1,j},$$

$$\hat{w}^l_{i,j}[k] = \frac{\exp(a_{i,j}[k])}{\sum_{k' \in \mathcal{K}_{i,j}} \exp(a_{i,j}[k'])}, \hat{e}^l_{i,j} = \sum_{k \in \mathcal{K}_{i,j}} \hat{w}^l_{i,j}[k]\hat{e}^l_{i,j}[k], a^l_{i,j} = \sum_{k \in \mathcal{K}_{i,j}} \hat{w}^l_{i,j}[k]a^l_{i,j}[k].$$

$$(1)$$

We elaborate on details of the $\text{Compose}$ function and the compatibility function $\phi$ afterwards. For a bottom cell at the first chart table $\mathcal{T}^1_{i,i}$, we initialize $\hat{e}^1_{i,i}$ as the embedding of $s_i$ and $a^1_{i,i}$ as zero. For $\mathcal{T}^0$, we assign a shared learn-able tensor to $\check{e}^0_{i,j}$. We encode the cells in the order of $\mathcal{B}$ so that when computing the representations and scores of a span, its inside pairs needed for the computation are already computed.

**Contextual Outside Pass.** A key difference in our outside pass is that we mix information from both inside and outside of a span. Meanwhile, to reduce the complexity of the outside to linear, we only consider cells and splits used in the inside pass. Analogous to inside, we denote the outside representation and score of a given span as $\check{e}^l_{i,j}[k]$ and $b^l_{i,j}[k]$ respectively, whose parent span is $(i, k)$

or $(k, j)$ for $k > j$ or $k < i$. Then, $\check{e}_{i,j}^l[k]$ and $b_{i,j}^l[k]$ can be computed by:

$$\check{e}_{i,j}^l[k] = \begin{cases} \text{Compose}_\beta^l(\hat{e}_{i,j}^l, \hat{e}_{j+1,k}^l, \check{e}_{i,k}^l) \\ \text{Compose}_\beta^l(\hat{e}_{k,i-1}^l, \hat{e}_{i,j}^l, \check{e}_{k,j}^l) \end{cases}, \bar{b}_{i,j}^l[k] = \begin{cases} \phi_\beta(\check{e}_{i,k}^l, \hat{e}_{j+1,k}^l) \\ \phi_\beta(\check{e}_{k,j}^l, \hat{e}_{k,i-1}^l) \end{cases}, b_{i,j}^l[k] = \begin{cases} a_{j+1,k}^l + \bar{b}_{i,j}^l[k] + b_{i,k}^l, & \text{for } k > j \\ a_{k,i-1}^l + \bar{b}_{i,j}^l[k] + b_{k,j}^l, & \text{for } k < i \end{cases} \quad (2)$$

The parameters of the Compose follow the order of left, right and parent span. According to $\mathcal{K}$ that records all valid splits for each span during the pruned inside pass, we can obtain a mapping from a span to its immediate sub-spans. By reversing such mapping, we get a mapping from a span to its valid immediate parent spans denoted as $\mathcal{P}$, which records the non-overlapping endpoint $k$ in the parent span $(i, k)$ or $(k, j)$ for a given span $(i, j)$. Thus, we have:

$$\check{w}_{i,j}^l[k] = \frac{\exp(b_{i,j}^l[k])}{\sum_{k' \in \mathcal{P}_{i,j}} \exp(b_{i,j}^l[k'])}, \check{e}_{i,j}^l = \sum_{k \in \mathcal{P}_{i,j}} \check{w}_{i,j}^l[k] \check{e}_{i,j}^l[k], b_{i,j}^l = \sum_{k \in \mathcal{P}_{i,j}} \check{w}_{i,j}^l[k] b_{i,j}^l[k]. \quad (3)$$

By following the reverse order of $\mathcal{B}$, we can ensure that when computing $\check{e}_{i,j}[k]$, the outside representation of parent span $(i, j)$ needed for the computation are already computed.[2]

**Composition and Compatibility functions.** We borrow the idea from R2D2 (Hu et al., 2021) to use a single-layered Transformer as the composition function. As shown in Figure 4, there are three special tokens indicating the role of each input. During the inside pass, we feed $\check{e}_{i,j}^{l-1}, \hat{e}_{i,k}^l, \hat{e}_{k+1,j}^l$ summed up with their role embeddings into the Transformer and take the output of $\check{e}_{i,j}^{l-1}$ as $\hat{e}_{i,j}^l[k]$ as shown in Figure 4(a). Analogously, we can get the outside representation as shown in Figure 4(b). To

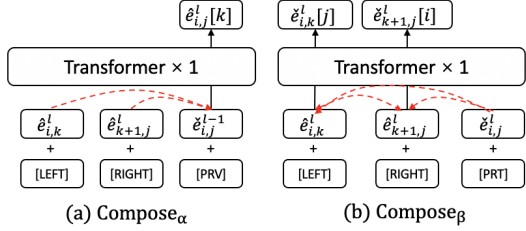

Figure 4: The Compose function for the inside and outside pass

compute the compatibility score, we feed $\hat{e}_{i,k}^l$ and $\hat{e}_{k+1,j}^l$ through $\text{MLP}_\alpha^L$, $\text{MLP}_\alpha^R$ used to capture left and right syntax features of sub-spans and calculate their scaled inner product:

$$\hat{u}_{i,k}^l = \text{MLP}_\alpha^L(\hat{e}_{i,k}^l), \hat{u}_{k+1,j}^l = \text{MLP}_\alpha^R(\hat{e}_{k+1,j}^l), \phi_\alpha(\hat{e}_{i,k}^l, \hat{e}_{k+1,j}^l) = (\hat{u}_{i,k}^l)^\intercal \hat{u}_{k+1,j}^l / \sqrt{d}, \quad (4)$$

where $u \in \mathbb{R}^d$. $\phi_\beta(\check{e}_{i,k}^l, \hat{e}_{j+1,k}^l)$ and $\phi_\beta(\check{e}_{k,j}^l, \hat{e}_{k,i-1}^l)$ can be computed in a similar way. For efficiency considerations, $\phi_\alpha$ and $\phi_\beta$ used in different layers share the same parameters.

**Tree induction.** For a given span $(i, j)$ at the $l$-th CIO layer, the best split point is $k$ with the highest $a_{i,j}^l[k], k \in \mathcal{K}_{i,j}$. Thus it is straightforward to recover the best parsing tree by recursively selecting the best split point starting from the root node $\mathcal{T}_{1,n}^L$ in a top-down manner, where $L$ is the total number of CIO layers.

**Multi-grained self-attention.** Once we recover the best parsing tree as described above (c.f., Figure 1(c)), we feed outside representations of nodes from the last CIO layer into subsequent Transformers, to enable constituents at different levels to communicate directly and output multi-grained representations fully contextualized with other spans.

### 3.2 PRE-TRAINING

---

**Algorithm 1** Pre-training ReCAT

1: **function** FORWARD($S, x, y$)
2:         ▷ $S$ is the original input, $x$ is the input with 15% tokens masked. $y$ is the target for MLM.
3:     $\mathcal{B} = \text{Prune}(S, \Phi)$     ▷ Corresponds to Section 3.1.1. $\Phi$ is the parameter for the top-down parser.
4:     Initialize $\mathcal{T}^0$     ▷ Initialize the chart table at the 0-th layer with word embeddings
5:     **for** $i \in 1$ to $L$ **do**     ▷ Run L-layered CIO blocks. Iteratively encode the sentence up and down.
6:         $\text{Inside}(\mathcal{B}, \mathcal{T}^i, \mathcal{T}^{i-1}, \alpha_i)$     ▷ Eq. 1, $\alpha_i$ is the inside parameters of the $i$-th CIO layer
7:         $\text{ContextualOutside}(\mathcal{B}, \mathcal{T}^i, \mathcal{T}^{i-1}, \beta_i)$     ▷ Eq. 2, $\beta_i$ is the outside parameters of the $i$-th CIO layer
8:     $z = \text{induce\_tree}(\mathcal{T}^L)$
9:     $r = \text{gather}(\mathcal{T}^L, z)$     ▷ Gather node representations from the last CIO layer.
10:     $\text{out} = \text{Transformers}(r)$     ▷ Feed contextualized node presentations into Transformers
11:     $\text{parser\_loss} = -\log p(z|S; \Phi)$     ▷ The loss for the top-down parser used in Prune.
12:     $\text{mlm\_loss} = \text{cross\_entropy}(\text{cls}(\text{out}), y)$     ▷ The loss for masked language modeling.
13:     **return** $\text{parser\_loss} + \text{mlm\_loss}$

---

[2]Detailed parallel implementation could be found in the Appendix A.1.

As our contextual outside pass brings information leakage, the original training objective designed for DIORA does not work anymore. Instead, here we use the MLM objective. For simplicity, we use the vanilla MLM objective instead of its more advanced variants such as SpanBERT (Joshi et al., 2020). Since each masked token corresponds to a terminal node in the parsing tree, we just take the Transformer outputs corresponding to these nodes to predict the masked tokens. As masking tokens may bring an extra difficulty to the top-down parser, we make all tokens visible to the parser during the pruning stage and only mask tokens during the encoding stage.

**Parser feedback.** We use a hard-EM approach (Liang et al., 2017) to optimize the parser used for the pruning. Specifically, we select the best binary parsing tree $\mathbf{z}$ estimated by the last CIO layer. As shown in Figure 2(e), at step $t$, the span corresponding to a given split $a_t$ is determined, which is denoted as $(i^t, j^t)$. Thus we can minimize the neg-log-likelihood of the parser as follows:

$$p(a_t|\mathbf{S}) = \frac{\exp(v_{a_t})}{\sum_{k=i^t}^{j^t-1} \exp(v_k)} \,, -\log p(\mathbf{z}|\mathbf{S}) = -\sum_{t=1}^{n-1} \log p(a_t|\mathbf{S}). \tag{5}$$

Algorithm 1 summarizes the procedure to pre-train ReCAT.

## 4 EXPERIMENTS

We compare ReCAT with vanilla Transformers and recursive models through extensive experiments on a variety of tasks.

**Data for Pre-training.** For English, we pre-train our model and baselines on WikiText103 (Merity et al., 2017). WikiText103 is split at the sentence level, and sentences longer than 200 after tokenization are discarded (about 0.04‰ of the original data). The total number of tokens left is 110M.

**ReCAT setups.** We prepare different configurations for ReCAT to conduct comprehensive experiments. We use ReCAT$_{share/noshare}$ to indicate whether the Compose functions used in the inside and outside passes share the same parameters or not. Meanwhile, we use ReCAT$[i, j, k]$ to denote Re-CAT made up of $i$ stacked CIO layers, a $j$-layer Compose function, and a $k$-layer Transformer. Specifically, ReCAT$_{share\ w/o\ iter}$ and ReCAT$_{share\ w/o\ TFM}$ are designed for the ablation study, corresponding to no iterative up-and-down mechanism and no Transformer layers.

| System | # params |
|---|---|
| Fast-R2D2 | 52M |
| Fast-R2D2+Transformer | 67M |
| DIORA*+Transformer | 77M |
| Parser+Transformer | 142M |
| Transformer×3 | 46M |
| Transformer×6 | 67M |
| Transformer×9 | 88M |
| ReCAT$_{noshare}[1, 1, 3]$ | 63M |
| ReCAT$_{share}[3, 1, 3]$ | 70M |
| ReCAT$_{noshare}[3, 1, 3]$ | 88M |
| ReCAT$_{noshare}[3, 1, 6]$ | 109M |
| ReCAT$_{share\ w/o\ iter}$ | 70M |

Table 1: Parameter sizes.

**Baselines.** For fair comparisons, we select baselines whose parameter sizes are close to ReCAT with different configurations. We list the parameter sizes of different models in Table 1. Details of baseline models are explained in the following sections.[3]

### 4.1 SPAN-LEVEL TASKS

**Dataset.** We conduct experiments to evaluate contextualized span representations produced by our model on four classification tasks, namely Named entity labeling (NEL), Semantic role labeling (SRL), Constituent labeling (CTL), and Coreference resolution (COREF), using the annotated OntoNotes 5.0 corpus (Weischedel et al., 2013). For all the four tasks, the model is given an input sentence and the position of a span or span pair, and the goal is to classify the target span(s) into the correct category. For NEL and CTL, models are given a single span to label, while for SRL and COREF, models are given a span pair.

**Baselines.** We use Transformer and Fast-R2D2 as our baselines. Fast-R2D2 could be regarded as a model only with the inside pass and using a 4-layer Transformer as the Compose function. For reference, we also include BERT as a baseline which is pre-trained on a much larger corpus than our model. For BERT, we use the HuggingFace (Wolf et al., 2019) implementation and select the cased version of BERT-base. Given a span, R2D2-based models (our model and Fast-R2D2) can encode the representation directly, while for Transformer-based models, we try both mean pooling and max pooling in the construction of the span representation from token representations.

**Fine-tuning details.** Our model and the Transformers are all pre-trained on Wiki-103 for 30 epochs and fine-tuned on the four tasks respectively for 20 epochs. We feed span representations through a two-layer MLP using the same setting as in Toshniwal et al. (2020). For the downstream

---

[3]Please find the hyper-parameters for the baselines in Appendix A.2.

| System | NEL | SRL | CTL | COREF |
|--------|-----|-----|-----|-------|
| Fast-R2D2 | 91.67/92.67 | 79.49/80.21 | 91.34/90.43 | 87.77/86.97 |
| Transformer6$_{mean}$ | 90.41/90.17 | 88.37/88.48 | 95.99/93.84 | 89.55/88.11 |
| Transformer6$_{max}$ | 90.49/90.70 | 88.86/88.99 | 96.33/95.28 | 90.01/89.45 |
| ReCAT$_{share}$[3, 1, 3] | **95.03/94.78** | **92.73/92.79** | **98.49/98.47** | **93.99/92.74** |
| Transformer9$_{mean}$ | 90.37/89.27 | 88.46/88.77 | 96.21/92.98 | 89.37/88.09 |
| Transformer9$_{max}$ | 90.93/90.28 | 89.00/89.12 | 96.49/95.92 | 90.32/89.38 |
| ReCAT$_{noshare}$[3, 1, 3] | **94.84/94.18** | **92.86/93.01** | **98.56/98.58** | **94.06/93.19** |
| **For Reference** | | | | |
| BERT$_{mean}$ | 96.49/95.77 | 93.41/93.49 | 98.31/97.91 | 95.63/95.58 |
| BERT$_{max}$ | 96.61/96.17 | 93.48/93.60 | 98.35/98.38 | 95.71/96.00 |

Table 2: Dev/Test performance for four span-level tasks on Ontonotes 5.0. All tasks are evaluated using F1 score. All models except BERT are pre-trained on wiki103 with the same setup.

classification loss, we use the cross entropy loss for single-label classification and binary cross entropy loss for multi-label classification. In particular, during the first 10 epochs of fine-tuning, inputs are also masked by 15% and the final loss is the summation of the downstream task loss, the parser feedback loss, and the MLM loss. For the last 10 epochs, we switch to the fast encoding mode, which is described in Appendix A.3, during which inputs are not masked anymore and the top-down parser is frozen. We use a learning rate of $5e^{-4}$ to tune the classification MLP, a learning rate of $5e^{-5}$ to tune the backbone span encoders, and a learning rate of $1e^{-3}$ to tune the top-down parser.

**Results and discussion.** Table 2 shows the evaluation results. We can see that ReCAT outperforms the vanilla Transformers across all four tasks, with absolute improvements of around 4 F1 scores on average. Furthermore, though BERT has twice as many layers as our model and is pre-trained on a corpus 20x larger than ours, ReCAT is still comparable with BERT (even better than BERT on the CTL task). The results verify our claim that explicitly modeling hierarchical syntactic structures can benefit tasks that require multiple levels of granularity, with span-level tasks being one of them. Since Transformers only directly provide token-level representations, all span-level representations reside in hidden states in an entangled form, and this increases the difficulty of capturing intra-span and inter-span features. On the other hand, our model applies multi-layered self-attention layers on the disentangled contextualized span representations. This enables the construction of higher-order relationships among spans, which is crucial for tasks such as relation extraction and coreference resolution. In addition, we also observe that although Fast-R2D2 works well on NEL, its performance in other tasks is far less satisfactory. Fast-R2D2 performs particularly poorly in the SRL task, where our model improves the most. This is because Fast-R2D2 solely relies on local inside information, resulting in a lack of spatial and contextual information from the outside which is crucial for SRL. On the other hand, our model compensates for this deficiency by contextualizing the representations in an iterative up-and-down manner.

### 4.2 SENTENCE-LEVEL TASKS

**Dataset.** We use the GLUE (Wang et al., 2019) benchmark to evaluate the performance of our model on sentence-level language understanding tasks. The General Language Understanding Evaluation (GLUE) benchmark is a collection of diverse natural language understanding tasks.

**Baselines.** We select Fast-R2D2, a DIORA variant, and vanilla Transformers as baselines which are all pre-trained on Wiki-103 with the same vocabulary. The Transformer baselines include Transformers with 3,6, and 9 layers corresponding to different parameter sizes of ReCAT with different configurations. [4] To study the gain brought by representations of non-terminal nodes, we include a baseline in which representations of non-terminal nodes in ReCAT are not fed into the subsequent Transformer, denoted as ReCAT$^{w/o\_NT}$. The settings of the Fast-R2D2 and the Transformers are the same as in Section 4.1. For reference, we list results of other RvNN variants such as Gumbel-Tree Choi et al. (2018), OrderedMemory (Shen et al., 2019a), and CRvNN (Chowdhury & Caragea, 2021).

**Fine-tuning details.** Our model and the Transformer models are all pre-trained on Wiki-103 for 30 epochs and fine-tuned for 20 epochs. Regarding to the training scheme, we use the same masked language model warmup as in 4.1 for models pre-trained via MLM. The batch size for all models is

---

[4]Detail setup of baselines is described in Appendix A.4

| System | natural language inference | | | single-sentence | | sentence similarity | |
|---|---|---|---|---|---|---|---|
| | MNLI-(m/mm) | RTE | QNLI | SST-2 | CoLA | MRPC(f1) | QQP |
| Fast-R2D2 | 69.64/69.57 | 54.51 | 76.49 | **90.71** | **40.11** | **79.53** | 85.95 |
| Fast-R2D2+Transformer | 68.25/67.30 | 55.96 | 76.93 | 89.10 | 36.06 | 78.09 | **87.52** |
| DIORA*+Transformer | 68.87/68.35 | 55.56 | 77.23 | 88.89 | 36.58 | 78.87 | 86.69 |
| Parser+Transformer | 67.95/67.16 | 54.74 | 76.18 | 88.53 | 18.32 | 77.56 | 86.13 |
| Transformer$\times 3$ | 69.20/69.90 | 53.79 | 72.91 | 85.55 | 30.67 | 78.04 | 85.06 |
| Transformer$\times 6$ | 73.93/73.99 | **57.04** | 79.88 | 86.58 | 36.04 | 80.80 | 86.81 |
| ReCAT$_{\text{noshare}}[1,1,3]$ | 72.77/73.59 | 54.51 | 73.83 | 84.17 | 23.13 | 79.24 | 85.02 |
| ReCAT$_{\text{share w/o iter}}$ | 75.03/75.32 | 56.32 | 80.96 | 84.94 | 20.86 | 78.86 | 85.36 |
| ReCAT$_{\text{share w/o NT}}$ | 74.24/74.06 | 55.60 | 80.10 | 85.89 | 28.36 | 79.03 | 85.98 |
| ReCAT$_{\text{share w/o TFM}}$ | 68.87/68.24 | — | — | 83.94 | — | — | — |
| ReCAT$_{\text{share}}[3,1,3]$ | **75.48/75.43** | 56.68 | **81.70** | 86.70 | 25.11 | 79.45 | 86.10 |
| Transformer$\times 9$ | 76.01/76.47 | 56.70 | **83.20** | 86.92 | **36.89** | 79.71 | **88.18** |
| ReCAT$_{\text{noshare}}[3,1,3]$ | 75.75/75.79 | **57.40** | 82.01 | 86.80 | 26.69 | **80.65** | 85.97 |
| ReCAT$_{\text{noshare}}[3,1,6]$ | **76.33/77.12** | 56.68 | 82.04 | **88.65** | 35.09 | 80.62 | 86.82 |
| **For reference** | | | | | | | |
| GumbelTree[†] | 69.50/ — | — | — | 90.70 | — | — | — |
| CRvNN[†] | 72.24/72.65 | — | — | 88.36 | — | — | — |
| Ordered Memory[†] | 72.53/73.2 | — | — | 90.40 | — | — | — |

Table 3: Evaluation results on GLUE benchmark. The models with † are based on GloVe embeddings and their results are taken from Ray Chowdhury & Caragea (2023). The others are pre-trained on wiki103 with the same setups.

128. As the test dataset of GLUE is not published, we fine-tune all models on the training set and report the best performance (acc. by default) on the validation set.

**Results and discussion.** Table 3 reports evaluation results on GLUE benchmark. First, we observe an interesting phenomenon that ReCAT significantly outperforms RvNN-based baselines on NLI tasks, underperforms the models on single-sentence tasks, and achieves comparable performance with them on sentence similarity tasks. Our conjecture is that the phenomenon arises from the contextualization and information integration abilities of the self-attention mechanism that enable the MLPs in pre-trained Transformers to retrieve knowledge residing in parameters in a better way. Therefore, for tasks requiring inference (i.e., knowledge beyond the literal text), ReCAT can outperform RvNN-based baselines. RvNN-based baselines perform well on tasks when the literal text is already helpful enough (e.g., single-sentence tasks and sentence similarity tasks), since they can sufficiently capture the semantics within local spans through explicit hierarchical composition modeling and deep bottom-up encoding. ReCAT, on the other hand, sacrifices such a feature in exchange for the contextualization ability that enables joint pre-training with Transformers, and thus suffers from degradation on single-sentence tasks. Second, even with only one CIO layer, ReCAT$_{\text{noshare}}[1,1,3]$ significantly outperforms Transformer$\times 3$ on the MNLI task. When the number of CIO layers increases to 3, ReCAT$_{\text{noshare}}[3,1,3]$ even outperforms Transformer $\times 6$. The results verify the effectiveness of explicit span representations introduced by the CIO layers. An explanation is that multi-layer self-attention over explicit span representations is capable of modeling intra-span and inter-span relationships, which are critical to NLI tasks but less beneficial to single-sentence and sentence similarity tasks. Third, the iterative up-and-down mechanism is necessary, as ReCAT$_{\text{noshare}}[3,1,3]$ significantly outperforms ReCAT$_{\text{share w/o iter}}$ and ReCAT$_{\text{noshare}}[1,1,3]$, even though ReCAT$_{\text{noshare}}[3,1,3]$ and ReCAT$_{\text{share w/o iter}}$ have almost the same number of parameters. We speculate that multiple CIO layers could enhance the robustness of the masked language models because the masked tokens may break the semantic information of inputs and thus affect the induction of syntactic trees, while multi-layer CIO could alleviate the issue by contextualization during multiple rounds of bottom-up and top-down operations. Finally, we notice that ReACT$_{\text{noshare}}[3,1,3]$ has a parameter size close to Transformer$\times 9$ but underperforms Transformer$\times 9$. This may stem from the difference in the number of layers (i.e., 6 v.s. 9). The results imply that the increase in parameters of the independent Compose functions in the inside and outside pass cannot reduce the gap brought by the number of layers. It is noteworthy that Parser+Transformer is worse than Fast-R2D2+Transformer, especially on CoLA, indicating that syntactic information learned from a few human annotations is less useful than that obtained by learning from vast amounts of data.

## 4.3 STRUCTURE ANALYSIS

We further evaluate our model on unsupervised grammar induction and analyze the accuracy of syntactic trees induced by our model.

| Model | mem. cplx | PTB $F_1(\mu)$ |
|---|---|---|
| Fast-R2D2$_{m=4}$ | $O(n)$ | 57.22 |
| ReCAT$_{share}[3,1,3]_{m=4}$ | $O(n)$ | 56.07 |
| ReCAT$_{share}[3,1,3]_{m=2}$ | $O(n)$ | 55.11 |
| ReCAT$_{noshare}[3,1,3]_{m=4}$ | $O(n)$ | **65.00** |
| ReCAT$_{noshare}[3,1,3]_{m=2}$ | $O(n)$ | 64.06 |
| ReCAT$_{noshare}$ w/o iter m=2 | $O(n)$ | 45.20 |
| For Reference | | |
| C-PCFG | $O(n^3)$ | 55.2† |
| NBL-PCFG | $O(n^3)$ | 60.4† |
| TN-PCFG | $O(n^3)$ | 64.1† |
| ON-LSTM | $O(n)$ | 47.7‡ |
| S-DIORA | $O(n^3)$ | 57.6† |
| StructFormer | $O(n^2)$ | 54.0‡ |

| Model | NNP | VP | NP | ADJP |
|---|---|---|---|---|
| Fast-R2D2 | 83.44 | 63.80 | 70.56 | 68.47 |
| ReCAT$_{share}[3,1,3]_{m=2}$ | 77.22 | 67.05 | 66.53 | 69.44 |
| ReCAT$_{share}[3,1,3]_{m=4}$ | **85.41** | 66.14 | 68.43 | 72.92 |
| ReCAT$_{noshare}[3,1,3]_{m=2}$ | 80.36 | 64.38 | 80.93 | **80.96** |
| ReCAT$_{noshare}[3,1,3]_{m=4}$ | 81.71 | **70.55** | **82.94** | 78.28 |

Table 4: Left table: F1 score of unsupervised parsing on PTB dataset. Values with † and ‡ are taken from Yang et al. (2022) and Shen et al. (2021) respectively.
Upper table: Recall of constituents.
Word-level: NNP (proper noun). Phrase-level: VP (Verb Phrase), NP (Noun Phrase), ADJP (Adjective Phrase).
Samples of deduced trees can be found in Appendix A.7.

**Baselines & Datasets.**  For fair comparison, we select Fast-R2D2 pre-trained on Wiki-103 as the baseline. In both models, only the top-down parser after training is utilized during test. Some works have cubic complexity, and thus not feasible to pre-train over a large corpus. Therefore, we only report their performance on PTB (Marcus et al., 1993) for reference. These works include approaches specially designed for grammar induction based on PCFGs such as C-PCFG (Kim et al., 2019a), NBL-PCFG (Yang et al., 2021), and TN-PCFG (Yang et al., 2022), and approaches based on language modeling tasks to extract structures from the encoder such as S-DIORA (Drozdov et al., 2020), ON-LSTM (Shen et al., 2019b), and StructFormer (Shen et al., 2021).

**Training details.**  We pre-train our model on Wiki-103 for 60 epochs, and then continue to train it on the training set of PTB for another 10 epochs. We randomly pick 3 seeds during the continue-train stage, then select the checkpoints with the best performance on the validation set, and report their mean f1 on the test set. As our model takes word pieces as inputs, to align with other models with word-level inputs, we provide our model with word-piece boundaries as non-splittable spans.

**Results and discussions.**  Table 4 reports the evaluation results. ReCAT significantly outperforms most of the baseline models in terms of F1 score and achieves comparable performance with TN-PCFG which is specially designed for the task. Moreover, from the cases shown in Appendix A.7, one can see strong consistency between the structures induced by ReCAT and the gold trees obtained from human annotations. The results indicate that the CIO layers can recover the syntactic structures of language, even though the model is learned through a fully unsupervised approach. The induced structures also greatly enhance the interpretability of Transformers since each tensor participating in self-attention corresponds to an explicit constituent. We also notice a gap between ReCAT$_{share}$ and ReCAT$_{noshare}$. To further understand the difference, we compute the recall of different constituents, as shown in the right table in Table 4. We find that both models work well on low-level constituents such as NNP, but ReCAT$_{noshare}$ is more performant on high-level constituents such as ADJP and NP. Considering the intrinsic difference in the meaning of the Compose function for the inside and outside passes, non-shared Compose functions may cope with such discrepancy in a better way. Despite of this, when referring to Table 3, we do not see a significant difference between ReCAT$_{share}$ and ReCAT$_{noshare}$ on downstream tasks. A plausible reason might be that the self-attention mechanism mitigates the gap owing to the structural deficiencies in its expressive power. The pruning threshold $m$ only has a limited impact on the performance of the model on grammar induction, which means that we can largely reduce the computational cost by decreasing $m$ without significant loss on the learned structures.

## 5 CONCLUSION & LIMITATION

We propose ReCAT, a recursive composition augmented Transformer that combines Transformers and explicit recursive syntactic compositions. It enables constituents at different levels to communicate directly, and results in significant improvements on span-level tasks and grammar induction.

The application of ReCAT could be limited by its computational load during training. A straightforward method is to reduce the parameter size of CIO layers. Meanwhile, the limitation can be mitigated to a significant extent under the fast encoding mode described in Appendix A.3 during the fine-tuning and the inference stage, whose total cost is around $2 \sim 3\times$ as the cost of Transformers.

## 6 ACKNOWLEDGEMENT

This work was supported by Ant Group through the CCF-Ant Research Fund.

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

# A APPENDIX

## A.1 PARALLELED IMPLEMENTATION FOR OUTSIDE

Unlike the inside pass in which the count of the inside pairs for a given span is $m$ at most, the count of the outside pairs for a given span is not determined. Thus this brings extra difficulty to compute weighted outside representations. To compute outside representations efficiently, we propose a cumulative update algorithm. We traverse span batches in the reverse order of $\mathcal{B}$ and then compute the outside representations for their immediate sub-spans and update their weighted outside representations by cumulative softmax as follows:

$$
\begin{bmatrix} \bar{w}_{i,k}^t \\ w_{i,k}^t \end{bmatrix} = \text{Softmax}\left( \begin{bmatrix} bcum_{i,k}^{l,t-1} \\ b_{i,k}^{l,t}[\widehat{p}] \end{bmatrix} \right), bcum_{i,k}^{l,t} = \log(\exp(bcum_{i,k}^{l,t-1}) + \exp(\bar{b}_{i,k}^{l,t}[\widehat{p}]))
$$
$$
\check{c}_{i,k}^{l,t} = \bar{w}_{i,k}^t \check{c}_{i,k}^{l,t-1} + w_{i,k}^t \check{c}_{i,k}^l[\widehat{p}], b_{i,k}^{l,t} = \bar{w}_{i,k}^t b_{i,k}^{l,t-1} + w_{i,k}^t b_{i,k}[\widehat{p}]
$$

(6)

As a sub-span may be updated many times, we denote the outside score at $t$-th time as $\bar{b}_{i,k}^{l,t}[\widehat{p}]$. We initialize $bcum_{i,k}^{l,0}$ as $-\infty$. Thus once all parents are traversed, the final result of $\check{c}_{i,k}^{l,t}$ will be equivalent to the result computed according to Equation 3.

## A.2 HYPERPARAMETERS.

For all baselines, we use the same vocabulary and hyperparameters but different layer numbers. We follow the setting in Devlin et al. (2019), using 768-dimensional embeddings, a vocabulary size of 30522, 3072-dimensional hidden layer representations, and 12 attention heads as the default setting for the Transformer. The top-down parser of our model uses a 4-layer bidirectional LSTM with 128-dimensional embeddings and a 256-dimensional hidden layer. The pruning threshold $m$ is set to 2. Training is conducted using Adam optimization with weight decay using a learning rate of $5 \times 10^{-5}$ for the tree encoder and $1 \times 10^{-3}$ for the top-down parser. Input tokens are batched by length, which is set to 10240. We pre-train all models on 8 A100 GPUs.

## A.3 FAST ENCODING

After pre-training, one can exploit ReCAT to encode a sequence of tokens. Though the complexity of the inside-outside process is $O(n)$ with $n$ the length of the sequence, the computational load is still many times higher than that of the vanilla Transformer. Specifically, the complexity of a single CIO layer is $O(m^2 n)$ with $m$ the pruning threshold (set as 2 in our experiments). Please refer to Appendix A.6 for detailed efficiency analysis. To further speed up encoding in fine-tuning and evaluation, we follow the force encoding proposed in Fast-R2D2. Specifically, we directly use the pretrained top-down parser to predict a parse tree for a given sentence and apply the trained Compose functions to compute contextualized representations of nodes following the tree and feed them into the Transformer.

## A.4 BASELINE SETUPS

The DIORA variant uses the linear inside-outside and a Transformer-based Compose function just as our model but still with the information access constraint. We also include an RvNN variant with a trained parser Zhang et al. (2020). The parser provides constituency trees and we use the encoder of Fast-R2D2 to compose constituent representations following the tree structure. The parser is trained with gold trees from Penn Treebank(PTB) (Marcus et al., 1993). Except for the RvNN+Parser baseline, all other models have no access to gold trees. For fair comparisons, we further combine Fast-R2D2, DIORA, and RvNN+Parser with Transformer by taking their node representations as the input of the Transformer and name the models Fast-R2D2+Transformer, DIORA+Transformer, and Parser+Transformer respectively.

### A.5 IMPROVED PRUNING ALGORITHM

We propose an improved implementation of the pruning algorithm which could reduce the steps to complete the inside algorithm from n to approximate $\log(n)$. Such an improvement could largely increase the training efficiency under parallel computing environments like GPU groups. The key idea is to simultaneously encode all cells whose sub-span representations are ready, thus more cells could be encoded in one batch even if they are not on the same row. To achieve this goal, we just need to build the encoding order of the cell batches according to the merge order, which is shown in Algorithm 2. Once the cell dependency graph is built, the cells at the bottom row are set as ready. Once a cell is set to ready, it will notify the higher-level cells that depend on it. Once all sub-spans of a cell are ready, the cell will be encoded in the next batch.

---

**Algorithm 2** Build cell batches

---

1: **function** BUILD_CELL_DEPENDENCIES($\mathcal{B}, \mathcal{T}$)
2:         $\triangleright$ $\mathcal{B}$ is the encoding order after pruning described in Section 3.1.1. $\mathcal{T}$ is the chart table.
3:  **for** $(i, j) \in \mathcal{B}$ **do**
4:   **for** $k \in \mathcal{K}_{i,j}$ **do**                $\triangleright$ $k$ is splits for span $(i, j)$.
5:    append($\mathcal{T}_{i,k}.\mathcal{N}, (i, j)$)    $\triangleright$ Add $(i, j)$ to the notify list. Once $(i, k)$ is encoded, inform $(i, j)$.
6:    append($\mathcal{T}_{k+1,j}.\mathcal{N}, (i, j)$)
7:
8: **function** BUILD_CELLS_BATCH($\mathcal{B}, \mathcal{T}$)
9:  BUILD_CELL_DEPENDENCIES($\mathcal{B}, \mathcal{T}$)
10:  $\mathcal{B} = []$                   $\triangleright$ Rebuild new encoding orders of cells
11:  $\mathcal{Q} = []$
12:  $\mathcal{Q}_{next} = []$
13:  **for** $i \in 0$ to $\mathcal{T}.length$ **do**
14:   $\mathcal{T}_{i,i}.ready = true$           $\triangleright$ Ready is default false for each cell.
15:   append($\mathcal{Q}, \mathcal{T}_{i,i}$)
16:  **for** $c \in \mathcal{Q}$ **do**             $\triangleright$ Iterate ready cells in the last batch
17:   **for** $(i, j) \in c.\mathcal{N}$ **do**
18:    **if** all sub-spans of $\mathcal{T}_{i,j}$ are ready & $|T_{i,j}.\mathcal{N}| > 0$ **then**
19:    $\triangleright$ If a cell does not participate in the encoding of higher-level cells, there is no need to encode it.
20:     $\mathcal{T}_{i,j}.ready = true$
21:     append($\mathcal{Q}_{next}, \mathcal{T}_{i,j}$)
22:   append($\mathcal{B}, \mathcal{Q}_{next}$)      $\triangleright$ Cells to encode for a single step in the inside-outside algorithm
23:   $\mathcal{Q} = \mathcal{Q}_{next}$
24:  **return** $\mathcal{B}$

---

### A.6 EFFICIENCY ANALYSIS

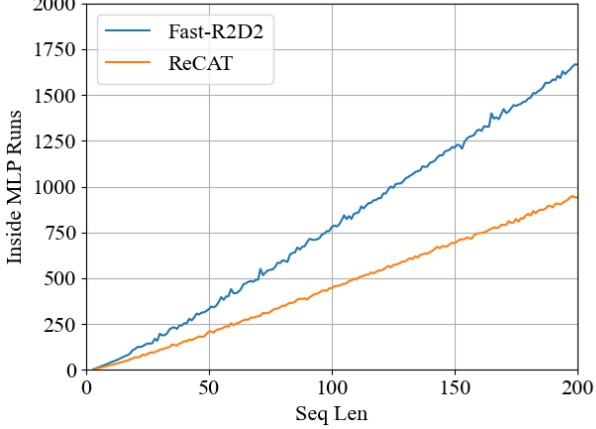

Figure 5: MLP runs for the inside pass.

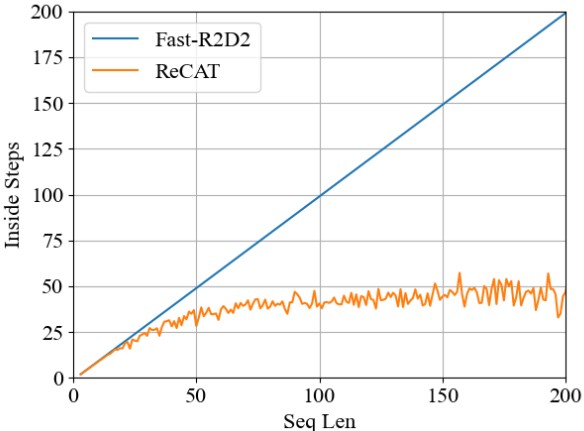

Figure 6: Steps for the inside pass.

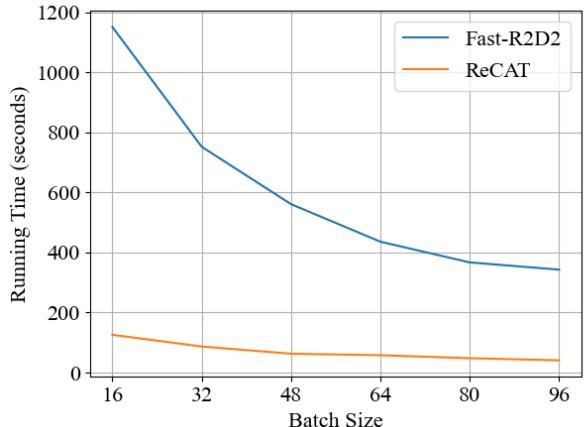

Figure 7: Practical training time.

To further study the efficiency improvement compared with the original pruning algorithm, we compare the MLP runs, inside steps, and training time of CIO layers using the original pruning algorithm versus our improved version. We set the pruning threshold $m$ to 2 and limit the input sentence length ranging from 0 to 200. We keep the model architecture the same but only differ in the pruning algorithm. We denote the original pruning algorithm as Fast-R2D2 and the improved one as ReCAT. The total training time is evaluated on 5,000 samples with different batch sizes on 1*A100 GPU with 80G memory. The result is shown in Figure 5, Figure 6 and Figure 7. The overall FLOPS is reduced by around 1/3. As we can find from the result, the total training time has a significant improvement.

### A.7   Samples of learned constituency trees

ReCAT:

The Dow Jones Industrial Average tumbled 39.55 points to 2613.73 in active trading

gold tree:

The Dow Jones Industrial Average tumbled 39.55 points to 2613.73 in active trading

ReCAT:

The bleak automotive results were offset by strong earnings from some non-automotive operations

gold tree:

The bleak automotive results were offset by strong earnings from some non-automotive operations

ReCAT:

Merrill Lynch Capital Markets Inc. is the sole underwriter for the offering

gold tree:

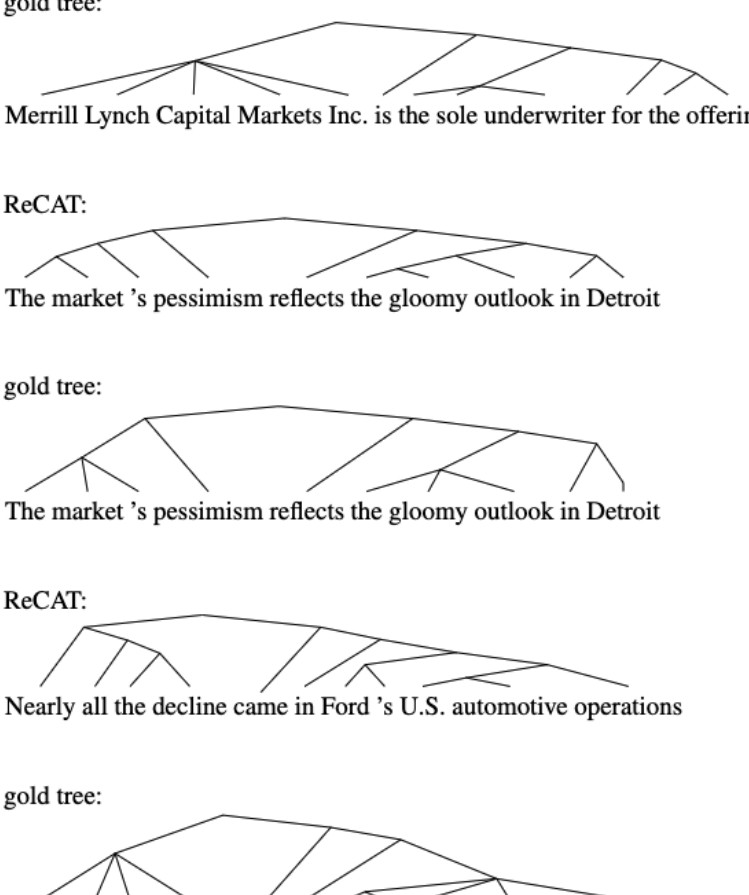

Merrill Lynch Capital Markets Inc. is the sole underwriter for the offering

ReCAT:

The market 's pessimism reflects the gloomy outlook in Detroit

gold tree:

The market 's pessimism reflects the gloomy outlook in Detroit

ReCAT:

Nearly all the decline came in Ford 's U.S. automotive operations

gold tree:

Nearly all the decline came in Ford 's U.S. automotive operations

