# OpenReview forum: "Augmenting Transformers with Recursively Composed Multi-grained Representations"
_ICLR.cc/2024/Conference — ICLR 2024 poster_

### Official Review · Reviewer_m86F · 2023-10-30

**Soundness:** 3 good
**Presentation:** 2 fair
**Contribution:** 2 fair
**Rating:** 6
**Confidence:** 4

**Summary:**

This paper proposes Recursive Composition Augmented Transformer (ReCAT), a model that aims to explicitly incorporate syntactic and structured trees into Transformers. The main change of ReCAT is the proposed contextual inside-outside (CIO) layers, which are inserted into a standard Transformer encoder right after the embedding layer. The CIO layer consists of the bottom-up and top-down modules. By stacking CIO layers, the iterative bottom-up and top-down layer-by-layer passes can unsupervisedly construct syntactic structures for the input sequence and produce contextualized representations for tree nodes.

By pre-trained with masked language modeling (MLM) and then fine-tuned, ReCAT was compared with Transformer-only methods and Fast-R2D2 and showed superior performance on multiple sentence-level and span-level tasks. Moreover, the syntactic trees induced by ReCAT exhibit strong consistency with human-annotated trees, proving that CIO layers can accurately learn syntactic structures.

**Strengths:**

- It is interesting and useful to augment transformers with multi-grained representations in an unsupervised manner.
- The proposed method is practical and effective based on the downstream task evaluation.

**Weaknesses:**

- The paper is poorly written, with many notations without explanation. Figure 2 is a copy from Figures in Fast-R2D2, with an unclear caption.
- Lack of important details, e.g. the computational cost of the proposed layers, how to choose the number of layers, and the hidden dimension of the stacked CIO layers.
- It would be helpful to explain the main differences and advantages of the proposed method compared with related works, e.g. Fast-R2D2.

**Questions:**

- What are the advantages of the proposed method compared with Fast-R2D2?
- What is the computational cost and complexity of the proposed layers?
- How to choose the hyperparameters and the model configurations of the ReCAT models?
- Can the proposed method be incorporated and augmented to pre-trained models like BERT?

---

> ### Author Response · Authors · 2023-11-17
> **Author response to reviewer m86F**
>
> Thank you for the time you spent reviewing our paper and for your valuable feedback.
>
> W.1: The paper is poorly written, with many notations without explanation. Figure 2 is a copy from Figures in Fast-R2D2, with an unclear caption.
>
> Thank you for reminding us of the clarity issues. We have gone through the paper again and explained some symbols that were not clarified before. To provide a better explanation of the entire pre-training process, we provided a pseudo-code in Algorithm 1.  We will keep improving the paper in terms of clarity.
>
> W.2: Lack of important details, e.g. the computational cost of the proposed layers, how to choose the number of layers, and the hidden dimension of the stacked CIO layers.
>
> Thanks for pointing out.
> The overall time complexity of CIO is $\mathcal{O}(m^2n)$, where n refers to sequence length and m refers to the pruning threshold(set to 2 as default).
> We also give a detailed analysis of computational cost in Appendix A.6.
> Moreover, we reduced the number of steps required to go through the entire CKY chart-table from $2n$ to $2\textrm{log}(n)$ and thus further reduced the training time on GPUs. Under the fast encoding mode, the overall computational cost is $2\sim3$ times vanilla Transformers.
> For a fair comparison, we choose $3$ as the number of the CIO layers and $768$ as the hidden dimension to make the size of the stacked CIO structure comparable to that of Fast-R2D2.
>
> W.3: It would be helpful to explain the main differences and advantages of the proposed method compared with related works, e.g. Fast-R2D2.
>
> Thanks for the suggestion. We have elaborated on the differences and the advantages in the general response to all reviewers. We also briefly explain the main differences and advantages of the proposed method in related works in the updated draft.
>
> Q.1. What are the advantages of the proposed method compared with Fast-R2D2?
>
> Please refer to the general response to all reviewers for details. In short, the major difference lies in the contextual outside pass which is not possessed by Fast-R2D2. Such a mechanism determines that the span representations in ReCAT are contextualized, i.e., with relative positional information and contextual information, and thus can be combined with Transformers. On the other hand, the span representations in Fast-R2D2 are span-local, lacking relative positional information and contextual information.
> Moreover, we also reduce the number of steps required to go through the entire CKY chart-table from $2n$ to $2\textrm{log}(n)$ which is detailed in Appendix A.5.
>
> Q.2. What is the computational cost and complexity of the proposed layers?
> Q.3 How to choose the hyperparameters and the model configurations of the ReCAT models?
>
> Please refer to our response regarding to W.2 above.
>
> Q.4. Can the proposed method be incorporated and augmented to pre-trained models like BERT?
>
> Yes, it can. A simple approach is to load the BERT parameters into the Transformer layers, and then jointly post-train the whole model. In practice, one may reduce the dimension of the CIO layers to reduce the computational cost but keep the feature of providing disentangled span

---

> ### Comment · Reviewer_m86F · 2023-12-04
>
> Sorry for the unclearness. The clarity of notations can be further improved by:
>
> (1) separating scalars and vectors, e.g. using bold to refer to vectors like representations $\hat{\mathbf{e}}\_{i,j}$ and using normal characters for scalars like scores.
>
> (2) Using different notations for different variables, e.g. the representations $\hat{\mathbf{e}}\_{i,j}$ and $\hat{\mathbf{e}}\_{i,j}[k] $ are not same, and $\hat{\mathbf{e}}\_{i,j}$ is the calculated from $\hat{\mathbf{e}}\_{i,j}[k]$ in Equation 1.
>
> (3) multiple equations can be split in lines, e.g. Equation 1,2,3.

---

### Official Review · Reviewer_fRFA · 2023-10-31

**Soundness:** 3 good
**Presentation:** 3 good
**Contribution:** 2 fair
**Rating:** 5
**Confidence:** 4

**Summary:**

This paper proposes contextual inside-outside (CIO) layers to explicitly model the syntactic structures of raw text in the encoding process of Transformers. The CIO layers are driven by a variant of the inside-outside algorithm proposed by DIORA (Drozdov et al. (2019) ). The authors leverage the linear neural inside algorithm proposed by R2D2 to prune cells that are unnecessary to encode during the bottom-up inside pass and propose to mix information from both inside and outside a span.

The benefits brought by CIO include (1) explicitly modeling syntactic structures without requiring parse tree annotations; (2) unlike previous work which only allows information flow within a span, CIO enables cross-span communications in a scalable way; (3) reducing the complexity of the inside-outside algorithm from cubic in DIORA to linear.

The major contributions are: (1) effectively combining ideas from two methods DIORA and R2D2 with innovative modifications. (2) introducing cross-span communication that breaks the information access constraint. (3) the experimental results on span-level tasks, natural language inference, and grammar induction demonstrate the effectiveness of the proposed method.

**Strengths:**

- Proposed an effective method to explicitly model the syntactic structure in the encoding process of transformers.

- It’s interesting to see that the Vanilla MLM can be used to replace the training objective of DIORA and the model can effectively learn the syntactic structure with the simple MLM objective.

- Results show that CIO can improve the transformer’s performance on span-based tasks and even help it to learn syntactic trees in an unsupervised manner.

**Weaknesses:**

- Contribution is not significant: the proposed method is an effective way of combining ideas of the inside-outside algorithm from DIORA and node-pruning algorithm from R2D2 with modifications to fit them into the transformer framework and address some weaknesses in DIORA and R2D2 such as the lack of communication between spans.

- Efficiency is low: (1) as mentioned by the author, the computational load is many times higher than that of the vanilla Transformer model and the complexity of a single CIO layer is O(m2n). (2) If using fast-r2d2, the CIO layer needs a pre-trained top-down parser to predict a parse tree for a given sentence.

- The major performance improvement is on Span-level tasks, and on sentence-level tasks ReCAT achieves worse performance compared with a vanilla transformer with similar size. On the structure-prediction task, ReCAT achieves similar performance compared with the previous method TN-PCFG.

**Questions:**

- How do you deal with sub-word tokens: if a word is broken into multiple tokens, how do you handle them in the CIO layer?

- In 3.3, you mentioned “Specifically, we directly use the pretrained top-down parser to predict a parse tree for a given sentence and apply the trained Compose functions”, does this mean that your method requires a pretrained top-down parser?

- Is it necessary to train the Transformer from scratch in the experiments? One way of using CIO layers is to use them as adaptors which can be plugged into multiple transformer layers. Training a model from scratch can impose significant computational cost and it could be more efficient to harness the pretraining of existing Transformers instead of training from scratch.

---

> ### Author Response · Authors · 2023-11-17
> **Author response to reviewer fRFA**
>
> Thank you for the time you spent reviewing our paper and for your valuable feedback.
>
> W.1: Contribution is not significant: the proposed method is an effective way of combining ideas of ..
>
> Yes, your understanding of our work is right. But beyond combining them, we further enable it to be jointly pre-trained with Transformers.
> The key weaknesses along the line of composition-based models are:
>
> (1) bad scalability
>
> (2) difficulty in combining with Transformers.
>
> We largely improve the scalability and demonstrate the key to combining with Transformers is joint pre-training. Meanwhile, joint pre-training is non-trivial for previous work like Fast-R2D2/DIORA. We explain the details in the general response.
>
> Combining composition-based models with Transformers is a meaningful direction as it allows for the integration of constituent-level symbolic interfaces. Currently, Language Models (LLMs) can engage in natural language conversations with humans. However, researchers face a limitation wherein they can only interact with Transformer models using tokens without a rich symbolic interface.
> Recent studies have demonstrated that utilizing spans as symbolic interfaces can yield fascinating outcomes. For instance, it enables alignment with visual information[1] and facilitates grounded lexicon learning[2]. Offering constituent-level interfaces for Transformers holds great potential. These interfaces can be combined with retrieval, graphs, and more, leading to the development of intriguing features for Transformers. The span-level task is also a good example. In ReCAT, each span has an explicit symbol interface to obtain its representation, and the results significantly outperform Transformers.
> However, it's still underexplored on how to integrate constituent-level symbolic interfaces with Transformers in an end-to-end approach, which is the key topic we discuss in this work.
>
> [1] Bo Wan, Wenjuan Han, Zilong Zheng, Tinne Tuytelaars. Unsupervised Vision-Language Grammar Induction with Shared Structure Modeling. ICLR 2022.
>
> [2] Jiayuan Mao, Haoyue Shi, Jiajun Wu, Roger P. Levy and Joshua B. Tenenbaum. Grammar-Based Grounded Lexicon Learning. NeurIPS 2021
>
> W.2: Efficiency is low.
>
> (1) ReCAT is indeed slower than vanilla Transformer,  but the CIO layer has been $5$ times faster than previous works like Fast-R2D2. We update the efficiency analysis in Appendix A.6. Moreover, we can control the cost via the fast encoding technique.
>
> (2) there might be misunderstanding regarding to ``if we need a pre-trained top-down parser''. In fact, the pre-trained top-down parser in fast encoding refers to the parser used in the CIO layers, which is pre-trained along with the CIO layers. Therefore, no extra pre-trained top-down parser is required. We improve the clarity by the pseudo-code (i.e., Algorithm 1) in the update of the draft.
>
> W.3:  About performance on sentence-level tasks compared with a vanilla transformer with similar size. On the structure-prediction task, ReCAT achieves similar performance compared with the previous method TN-PCFG.
>
> Yes, but we argue when the total layer numbers are the same, our results are still competitive on GLUE. We additionally add $ReCAT_{noshare}$[3,1,6] whose total layer number is 9. Its performance is competitive with Transformer 9.
> Regarding grammar induction, TN-PCFG is close to ReCAT in terms of performance, but considering its $O(n^3)$ time complexity while ReCAT is just $O(n)$, we think the advantage of ReCAT over TN-PCFG is huge in practice. Meanwhile, we can further improve via pre-training on the larger corpus.
>
> Q.1. How do you deal with sub-word tokens: if a word is broken into multiple tokens, how do you handle them in the CIO layer?
>
> Handling the sub-word tokens does not need any special operations. We just take them as ordinary input tokens. The CIO layers can learn how to compose them properly during pre-training.
>
> Q.2.  In 3.3, you mentioned “Specifically, we directly use the pre-trained top-down parser to predict a parse tree for a given sentence and apply the trained Compose functions”, does this mean that your method requires a pre-trained top-down parser?
>
> No and we have clarified this point in response to W.2.
>
> Q.3. Is it necessary to train the Transformer from scratch?
>
> We suppose it's necessary for a research paper.
> As there is no previous work about whether span representations are helpful to augment Transformers, we need to compare ReCAT and vanilla Transformers as fairly as possible.
> Meanwhile, since the pre-trained Transformer has never seen span representations, we still need to post-train ReCAT to adopt those new span representations from CIO layers. It costs more computational resources to post-train on a corpus like OpenWebText than from scratch on wiki-103. Moreover, it would be hard to tell if the gain is from extra training steps or CIO layers.

---

> ### Author Response · Authors · 2023-11-18
> **Further clarification**
>
> There might be a misunderstanding regarding "It’s interesting to see that the Vanilla MLM can be used to replace the training objective of DIORA and the model can effectively learn the syntactic structure with the simple MLM objective."
>
> Simply replacing the training objective of DIORA with the MLM objective $\textbf{doesn't work well}$.  In Table 3 and Table 4. $ReCAT_{noshare}[1,3,3]$  can be viewed as an extended version of DIORA (run inside-outside algorithm only once, with 3-layered Transformers as encoder, but replacing the outside pass with our contextual outside pass, pretrained via MLM), but it does not perform well on grammar induction and glue tasks. The original version is supposed to perform worse if pretrained via MLM.
>
> Fast-R2D2+DIORA+MLM doens't work well. Fast-R2D2+DIORA+ iterative up-and-down mechanism + contextual outside + MLM makes it work.  The iterative up-and-down mechanism(stackable CIO layers) and contextual outside is our key innovation,  enabling the CIO layer to be pre-trained through Masked Language Modeling, thereby making it possible to achieve joint pre-training with Transformer through MLM.
>
> Please note that the original inside-outside algorithm is not stackable, as the outside algorithm was originally designed to construct language modeling objectives, thus there is no interaction between information inside and outside of a span. However, our contextual outside pass is designed to contextualize span representations,  allowing for stacking and adapting to Masked Language Modeling.

---

### Official Review · Reviewer_46mR · 2023-11-01

**Soundness:** 4 excellent
**Presentation:** 3 good
**Contribution:** 3 good
**Rating:** 8
**Confidence:** 4

**Summary:**

The paper proposes a syntax-augmented transformer based on producing span representations following the inside and outside traversals of a parse chart. Unlike existing methods following this idea (Fast-R2D2, DIORA), they incorporate cross-span contextualization. The resulting architecture is competitive with vanilla transformers on span-level and GLUE tasks while achieving strong grammar induction results.

**Strengths:**

(1) The method performs well compared to both transformers and Fast-R2D2 on span-level and sentence-level tasks, while also achieving strong grammar induction results.

(2) The method is evaluated on a wide range of tasks and compared to a wide range of baselines.

**Weaknesses:**

(1) As the authors mention, one of the main motivations for augmenting transformers with syntax is to improve compositional generalization (as well as potentially interpretability and controllability). Therefore, while the method does outperform transformers for span-level tasks, I think the results would stronger if there were experiments addressing these original motivations.

(2) While I understand that the method is inherently complex, I still think the methods section could be made a little clearer. For example, maybe a self-contained step-by-step summary of the algorithm at the end of the section would be helpful.

(3) While their method does outperform Fast-R2D2, the ideas feel very similar, limiting the scientific contribution of this work. As it stands, it feels a bit like a complex combination of ideas from Fast-R2D2 and DIORA, where the takeaway is a bit unclear. Is the key difference scalability, the use of transformers, pruning, cross-span communication, or all of the above? If the key difference is cross-span communication, why is that important and what are the specific changes in the algorithm that enable this difference? And are there ablation experiments that support the claim that cross-span communication is important?

(4) Related to (3), the paper would benefit from ablations to support the claims of why certain design decisions were effective.

**Questions:**

(1) I find it a bit surprising that Fast-R2D2+Transformer is worse than Fast-R2D2 on GLUE (Table 3). Is there some intuition for why this is the case?

(2) Is there an ablation for ReCat without extra transformer layers?

(3) What are the model sizes for ours_{share} and ours_{noshare} in Table 2?

---

> ### Author Response · Authors · 2023-11-17
> **Author response to reviewer 46mR**
>
> Thank you for the time you spent reviewing our paper and for your valuable feedback.
>
> W.1: about compositional generalization.
>
> in addition to the span-level tasks, we also observed significant gains on the grammar induction task (i.e., Table 4 in the paper). This could be another piece of evidence that indicates the compositional generalization of the model.
> As compositional generalization refers to the ability to recognize or generate novel combinations of observed elementary concepts, our model has never seen sentences in the test set in the grammar induction task but induces their structures well.
> Besides, we also show a few parsing results from ReCAT in Appendix A.7, through which we aim to show what the syntax augmented Transformer really learned. Indeed, the showcases exhibit high consistency with human-annotated structures (i.e., the gold trees), meaning that the superior performance of ReCAT on the span-level tasks may stem from the recovery of the underlying syntactic structures of texts.
>
> W.2: ..., maybe a self-contained step-by-step summary of the algorithm at the end of the section would be helpful.
>
> We appreciate your suggestion and add pseudo code (i.e., Algorithm 1) in the updated draft to show how the model is pre-trained step-by-step.
>
> W.3: ...,the takeaway is a bit unclear
>
> We have discussed in detail about the advantages of ReCAT and the innovation we made in the work in the general response. The takeaway is
>
> (1) the key advantages of ReCAT over DIORA and Fast-R2D2 are the capability of joint pre-training with Transformer and scalability;
>
> (2) to achieve the advantages, we develop the iterative up-and-down mechanism (via stacking multiple CIO layers) and the contextual outside pass, which represent the key difference we made in the work comparing to DIORA and Fast-R2D2.
>
> Please note that the original inside-outside algorithm is not stackable, as the outside algorithm was originally designed to construct language modeling objectives while ours is designed to contextualize span representations, allowing for stacking and adapting to Masked Language Modeling.
>
> W.4: Related to (3), the paper would benefit from ablations to support the claims of why certain design decisions were effective.
>
> Thanks for the suggestion. Since the iterative up-and-down mechanism and the contextual outside pass are key technical contributions, it is better for us to ablate the model so as to inspect the effect of each design. In fact, we have done the ablation in the experiments:
>
>  (1) ablation of iterative up-and-down. This is done by the comparison between $ReCAT_{share}$[1,3,3] and $ReCAT_{share}[3,1,3]$ in Table 3. The two models have similar parameter sizes, but $ReCAT_{share}[3,1,3]$ contains 3 layers of CIO while $ReCAT_{share}[1,3,3]$ only has one, i.e, no iterative up-and-down.
> The results in Table 3 indicate that when the total numbers of parameters are close, stacking CIO layers can bring consistent improvement. Additionally, we update Table 4  by including $ReCAT_{share}[1,3,3]$. Again, the big gap between the two models indicates the value of the proposed mechanism.
> We additional added an explanation for them in the updated draft.
>
> (2) ablation of joint pre-training. This can be checked by comparing ReCAT$_{share}[3,1,3]$ with Fast-R2D2+Transformer and DIORA+Transformer in Table 3.  Transformers used in these baselines are randomly initialized.
> From the results, we can conclude that the joint pre-training is important.
>
> Response to Q1:
>
> We were surprised too when we first observed the results. Appending Transformers is helpful for QQP but leads to no gain for MNLI.
> And to find out ``why'' motivates the work. As we analyze in Section 4.2, composition-based models are good at tasks that only need literal
> text information while Transformers are good at tasks requiring information beyond literal texts, i.e., knowledge.
> We realized the importance of joint pretraining. Knowledge entailed in corpora is compressed into Transformers during the pretraining.
> Just appending Transformers without joint pre-training, the two are incompatible.
> However, as we have elaborated in the general response, one cannot perform joint pre-training by simply appending Transformers after Fast-R2D2 or DIORA. Thus we propose CIO layers to address a series of issues even including scalability.
>
> Response to Q2:
>
> We update Table 3 in the draft by adding a baseline ReCAT$_{share}[3,1,0]$ (i.e., no transformer layers). As expected, it is worse than Fast-R2D2 and DIORA. The comparison indicates that MLM itself is not so efficient as the original objective used by R2D2 and DIORA (15\% vs 100\%). The advantage of MLM lies in its ability to make pre-training with Transformer feasible.
>
> Response to Q3:
>
> Sorry for the abuse of notations. They are $ReCAT_{share}[3,1,3]$ and $ReCAT_{noshare}[3,1,3]$ respectively whose parameter sizes are shown in Table 1. We have updated the draft to improve clarity.

---

> > ### Comment · Reviewer_46mR · 2023-11-23
> >
> > Thanks for the reply and revisions. I still think the paper could be made clearer, but I have raised my score to an 8.

---

### Official Review · Reviewer_FCy4 · 2023-11-10

**Soundness:** 3 good
**Presentation:** 3 good
**Contribution:** 3 good
**Rating:** 6
**Confidence:** 3

**Summary:**

The paper proposed to augment the Transformer architecture with a contextual inside-outside layer (CIO). The CIO layer model explicitly the recursive syntactic compositions. The CIO layer goes in between the embedding layer and the self-attention layer. The author proposed a new variant of the inside-outside algorithm with contextualization ability. The CIO layer allows modeling of the inter-span and intra-span interactions. Experiments show that the proposed method outperforms the plain Transformer models on span-level tasks. It also outperforms recursive models on natural language inference tasks. The CIO also has good interpretability since it has strong consistency with human-annotated syntactic trees.

**Strengths:**

The author proposed a novel method that include the recursive syntactic composition information into the Transformer architecture. The method is verified to be effective, especially for span-based tasks, by experiments with

**Weaknesses:**

As authors pointed out, the CIO layer can be expensive. I'm also wondering how is the effect

**Questions:**

1. It would be interested to see if the gains still holds when we scale to larger models. (It is recognized it's not always possible/easy to experiment with a larger model.)
2. I'm wondering if the CIO layer can be trained at the finetuning stage so that we can still utilize a powerful pre-trained model and augment it with the CIO layer only at the finetuning stage.
3. This might be diverging or outside the scope of the paper, but I'm curious how tokenization would be affecting the effectiveness of the proposed method. For example, if it's byte or character-level vocabulary, do we observe a similar gain.

---

> ### Author Response · Authors · 2023-11-14
> **Dear Reviewer FCy4, may I have your complete comments?**
>
> Thanks for your insightful comments. But part of them seem incomplete, especially strengths and weaknesses. So could you kindly paste full of them once you have time? Many Thanks.
>
> We are still preparing the rest part of the authors' response. But the question about character-level parsing we can respond first.
> We did try applying the inside-outside algorithm on character-level vocabulary to see if it could induce underlying structures of characters. Unfortunately, it doesn't work well in languages based on spelling like English. A key reason we guess is the information entropy is too low as there are only 26 characters in English. Thus feedback is so weak for the inside-outside algorithm. However, it could work well in languages with large basic character-level vocabulary like Chinese.

---

> ### Author Response · Authors · 2023-11-17
> **Author response to reviewer FCy4**
>
> Thank you for the time you spent reviewing our paper and for your valuable feedback.
>
> W.1: As the authors pointed out, the CIO layer can be expensive. I'm also wondering how is the effect.
>
> As we have explained in the general response, with all the effort (contextual inside-outside with $\mathcal{O}(n)$ time complexity and $2\textrm{log}(n)$ steps), the CIO layer has been $5$ times faster than using the original pruning algorithm proposed in Fast-R2D2. We update the efficiency analysis in Appendix A.6 in the draft. On the other hand, training a ReCAT model is indeed more expensive than training a vanilla Transformer with a similar parameter size ($88M$) due to the dynamic programming in stacked CIO layers. We can control the running time of ReCAT to be about 2 to 3 times that of the vanilla Transformer during inference and fine-tuning using the fast encoding technique described in Appendix A.3. Considering the outstanding performance of ReCAT on span-level tasks and the grammar induction task, we think the cost is worthwhile.
>
> Q.1. It would be interested to see if the gains still holds when we scale to larger models.
>
> We train a larger ReCAT with 3 CIO layers and 6 Transformer layers (namely ReCAT$_{noshare}[3,1,6]$), and update the results in Table 3 in the draft. We can see that the performance of ReCAT continuously gets improved with the increase in model size ($[1,1,3] \rightarrow [3,1,3] \rightarrow [3,1,6]$).
>
> Q.2. I'm wondering if the CIO layer can be trained at the finetuning stage so that we can still utilize a powerful pre-trained model and augment it with the CIO layer only at the finetuning stage.
>
> Though we are not able to finish the extra experiments in such a short rebuttal period, we believe that joint post-training is still necessary to adapt to span-level representations, as the pre-trained model has never seen those span-level representations during its pre-training period.
>
> Q.3. this might be diverging or outside the scope of the paper, but I'm curious how tokenization would be affecting the effectiveness of the proposed method. For example, if it's byte or character-level vocabulary, do we observe a similar gain.
>
> Please refer to the previous reply.

---

### Author Response · Authors · 2023-11-17
**General response**

We appreciate the comments from all reviewers that help us improve the work.

1. Advantages over DIORA and Fast-R2D2

In the table below, we present a detailed comparison among CIO (the key component of ReCAT), DIORA, and Fast-R2D2 from five aspects including (1) P.W.T: if one can pre-train the model together with Transformer; (2) time cplx.: time complexity; (3)  arch.: architecture in terms of inside-outside or inside-only; (4) \#steps: number of steps required to go through the entire CKY chart-table; and (5) avg ctx. len: length of context used in the prediction of a token during pre-training.  We highlight P.W.T and scalability (related to time complexity and number of steps) as the key advantages over DIORA and Fast-R2D2 that also shape the contributions of the work.

|           | P.W.T. | time cplx. | arch.          | \#steps | avg ctx. len. |
|-----------|-------|------------|----------------|---------|---------------|
| DIORA     | No    | O($n^3$)     | inside-outside | 2n      | n             |
| Fast-R2D2 | No    | O(n)       | inside-only    | n       | 2m            |
| CIO       | $\textbf{Yes}$   | O(n)       | inside-outside | $\textbf{2log(n)}$ | n             |

Table 1. Comparison among CIO, DIORA, and Fast-R2D2. $n$ refers to the length of a sequence and $m$ refers to the pruning threshould

$\textbf{P.W.T.}$ It is not trivial to jointly pretrain DIORA and Fast-R2D2 with Transformers, because they have to strictly obey the information access constraint (i.e., information flow is only allowed within (inside) or among (outside) spans) to ensure their objectives.  Pre-training with Transformers brings information leak, which breaks the assumption of the objectives of the two models.

Instead of the original objectives, one may turn to MLM. However, the representations of Fast R2D2 and DIORA lack relative positional information, and the masked tokens will hinder the grammar induction, which makes the quick remedy fail to work. As a reference, one can check the performance of $ReCAT_{share}$[1,3,3] in Table 3 and Table 4. ReCAT$_{share}[1,3,3]$ is a ReCAT model without the iterative up-and-down mechanism (i.e., only one CIO layer), and it can be viewed as an extended version of DIORA (with contextual outside pass), but is much worse than ReCATs with stacked CIO layers (one of our innovations in the work).

One may argue if we really need "pre-training" with Transformer. Shall we just skip pre-training and initialize the input layer of Transformers with the output of DIORA or Fast-R2D2 for fine-tuning as a simple alternative of ReCAT? This straightforward method, however, is proven ineffective by the comparison with DIORA+Transformer and Fast-R2D2+Transformer in Table 3.

Therefore, it is necessary to enable joint pre-training if one aims to achieve the best of both worlds. This is exactly what we deal with in the work, and we achieve this through $\textbf{the iterative up-and-down mechanism}$ implemented via stacking multiple CIO layers (corresponding to lines 5-7 in Algorithm 1.) and the $\textbf{contextual outside pass}$ allowing information exchange (corresponding to Equation 2 and Figure 4 (b)), which represent the key technical innovations and let the work stand out in the research of syntax augmented models.
Thanks to the innovations, ReCAT can learn relative positional and contextual information, and improve robustness to masked input tokens, which gets joint pre-training with an MLM objective to work.

$\textbf{Scalability.}$ Due to the $\mathcal{O}(n^3)$ time complexity, DIORA is not able to scale up by nature.  Both ReCAT and Fast-R2D2 have linear time complexity in the inside pass, but Fast-R2D2 does not have an outside mechanism. We develop an outside algorithm with linear time complexity in ReCAT (corresponding to Section 3.1.2 Contextual Outside Pass). Hence, the overall time complexity of the CIO layer is $O(n)$ ($n$ refers to the length of a sequence), which already makes the model scale well. Moreover, we further enhance the training efficiency of ReCAT by reducing the number of steps required to go through the entire CKY chart-table from $2n$ to approximate $\sim 2\textrm{log}(n)$. By the technique, the training time of the CIO layers is reduced to about $1/5$ of the original duration in a GPU environment. We present the details of the technique in Appendix A.5 in the update of the draft.

2. Updates in our draft.

After submission of the original draft, we discovered that the pruning algorithm can be further optimized to complete the inside-outside algorithm in $2log(n)$ steps without affecting the original model structure. This can accelerate the training of the CIO layers. We have included this in the main body of the text and provided a detailed description of the algorithm in Appendix 5. We have also made the necessary adjustments to the paper based on the suggestions of the reviewers. The revised sections are highlighted in blue.

---

### Meta-Review · Area_Chair_maX6 · 2023-12-05

**Metareview:**

The paper introduces the Recursive Composition Augmented Transformer (ReCAT), incorporating a novel contextual inside-outside (CIO) layer into the Transformer architecture to model hierarchical syntactic structures without relying on predefined trees. This innovative approach allows for deep intra-span and inter-span interactions, significantly outperforming traditional Transformer models in various natural language tasks and demonstrating strong consistency with human-annotated syntactic trees, enhancing both performance and interpretability. While the method shows effectiveness in span-based tasks and natural language inference, it also presents challenges in computational expense and complexity. Overall, this paper represents a substantial contribution to the quest of searching for new architectures, making it a solid and impactful work.

**Justification For Why Not Higher Score:**

The architecture is still based on transformer.

**Justification For Why Not Lower Score:**

The work is generally solid.

---

### Decision · Program_Chairs · 2024-01-16

Accept (poster)